# shRNA-Targeting Caspase-3 Inhibits Cell Detachment Induced by Pemphigus Vulgaris Autoantibodies in *HaCaT* Cells

**DOI:** 10.3390/ijms25168864

**Published:** 2024-08-14

**Authors:** Deyanira Pacheco-Tovar, María-Guadalupe Pacheco-Tovar, Santiago Saavedra-Alonso, Pablo Zapata-Benavides, Felipe-de-Jesús Torres-del-Muro, Juan-José Bollain-y-Goytia, Rafael Herrera-Esparza, Cristina Rodríguez-Padilla, Esperanza Avalos-Díaz

**Affiliations:** 1Department of Immunology, School of Biological Sciences, UACB, Universidad Autónoma de Zacatecas, Av. de la Revolución Mexicana s/n, Colonia Tierra y Libertad, Guadalupe CP 98615, Zacatecas, Mexico; deyanirapachecotovar@uaz.edu.mx (D.P.-T.); lupitapacheco@uaz.edu.mx (M.-G.P.-T.); felipe.torresde@uanl.edu.mx (F.-d.-J.T.-d.-M.); jjbollainygoytia@uaz.edu.mx (J.-J.B.-y.-G.);; 2School of Chemistry Sciences, Universidad Autónoma de Zacatecas, Campus Universitario Siglo XXI, Carretera Zacatecas-Guadalajara, Ejido “La Escondida”, Zacatecas CP 98160, Zacatecas, Mexico; 3Department of Immunology and Virology, Faculty of Biological Sciences, Universidad Autónoma de Nuevo León, Av. Pedro de Alba s/n, Ciudad Universitaria, San Nicolás de los Garza CP 64450, Nuevo León, Mexico; ssaavedraln@uanl.edu.mx (S.S.-A.); cristina.rodriguezpd@uanl.edu.mx (C.R.-P.)

**Keywords:** pemphigus autoantibodies, shRNA, cell adhesion, acantholysis, caspase-3, apoptosis

## Abstract

Pemphigus is an autoimmune disease that affects the skin and mucous membranes, induced by the deposition of pemphigus IgG, which mainly targets desmogleins 1 and 3 (Dsg1 and 3). This autoantibody causes steric interference between Dsg1 and 3 and the loss of cell adhesion, producing acantholysis. This molecule and its cellular effects are clinically reflected as intraepidermal blistering. Pemphigus vulgaris-IgG (PV-IgG) binding involves p38MAPK-signaling-dependent caspase-3 activation. The present work assessed the in vitro effect of PV-IgG on the adherence of *HaCaT* cells dependent on caspase-3. PV-IgG induced cell detachment and apoptotic changes, as demonstrated by annexin fluorescent assays. The effect of caspase-3 induced by PV-IgG was suppressed in cells pre-treated with caspase-3-shRNA, and normal IgG (N-IgG) as a control had no relevant effects on the aforementioned parameters. The results demonstrated that shRNA reduces caspase-3 expression, as measured via qRT-PCR and via Western blot and immunofluorescence, and increases cell adhesion. In conclusion, shRNA prevented in vitro cell detachment and the late effects of apoptosis induced by PV-IgG on *HaCaT* cells, furthering our understanding of the molecular role of caspase-3 cell adhesion dependence in pemphigus disease.

## 1. Introduction

Pemphigus is an autoimmune disease characterized by the formation of intraepidermal blisters on the skin and/or mucous membranes that are triggered by accumulated IgG autoantibodies that target the desmosomal proteins of keratinocytes, resulting in the loss of intercellular junctions and acantholysis. There are different clinical forms of pemphigus which histologically correlate with the molecular and structural expression of Dsg1 or Dsg3 in the epidermis and mucous membranes [1,2,3].

The main mechanism of PV pathology is the direct steric interference of homophilic and heterophilic Dsg:Dsg and Dsc:Dsg binding [4,5,6]. All other mechanisms, including p38MAPK-signaling dependence and apoptosis, are probably an epiphenomenon of the direct steric effects of PV-IgG binding, and it has been proposed that acantholytic cells may develop cell death as their ultimate fate. This initial assumption of apoptosis in pemphigus was mainly based on TUNEL assay results and other individual techniques used to disclose apoptosis, suggesting that the extrinsic pathway or cell death receptors of the tumor necrosis factor (TNF) family are involved in pemphigus pathology, including Fas (CD95) and its cognate ligands, which can be found in blisters or their vicinity. A similar role has been attributed to the epidermal growth factor receptor (EGFR); however, even accounting for recent advances, the role of apoptosis in pemphigus is still under investigation [7,8,9,10].

The present study focused on inhibiting caspase-3, which we hypothesized would reduce cell detachment and acantholysis. Our study was based on results obtained by other investigators in a knockout mouse model that demonstrated a lack of keratinocyte fragility and/or blistering amelioration in keratinocyte-specific caspase-3-deficient mice (casp-3EKO) injected with a potent anti-Dsg3-specific antibody. Additionally, our experimental data and the results of others support the possible role of caspase-3 in acantholysis, as the inhibition of caspase-3 by Ac-DEVD-cmk prevents the formation of intraepidermal blisters in neonatal BALB/c mice injected with PV-IgG [11,12,13,14]. The aforementioned results suggest that inhibiting caspase-3 activity is an attractive mechanism for partially understanding the pathophysiology of pemphigus in terms of cell adherence. The effect of caspase-3 in PV has been studied extensively, but our approach has the potential to further address the downstream markers of adhesion affected by PV. This is important because cell detachment is a key mechanical point in blister formation, and this is not fully understood. Thus, the present study further analyzed whether suppressing caspase-3 mRNA with shRNA modifies cell adhesion, decreases cell death, and affects keratinocyte proliferation in vitro.

## 2. Results

### 2.1. PV-IgG

PV-IgG was isolated from untreated patient plasma with clinical, serologic, and histologic confirmations of the disease. This sample was taken during the clinical phase of disease activity with an anti-epithelial antibody titer of 1:2560 (negative value < 1:160), an anti-Dsg1 level of 254 UR/mL, and an anti-Dsg3 level of 281 UR/mL (negative value < 20). PV-IgG was purified through affinity chromatography to preserve its anti-epithelial and anti-Dsg1 and 3 activities. Normal IgG (N-IgG) was negative for anti-epithelial and anti-desmoglein 1 and 3 activities (Figure 1a,b).

### 2.2. Casp-3-shRNA Expression in Transformants

Expression vectors (Figure 2a) that propagated into competent cells were isolated and characterized via electrophoresis, and positive clones appeared as linear plasmids (Figure 2b).

### 2.3. Effect of shRNA on Caspase-3 of Transfected Cells

Positive transformants were used to transfect *HaCaT* cells, and the transfection efficiency was 72%. Then, the effect of the short hairpin on caspase-3 mRNA was tested using a qRT-PCR assay to ensure that caspase-3 expression was inhibited. shRNA reduced caspase-3 mRNA expression, which, under basal conditions, was at a level similar to that in cells treated with N-IgG. However, with PV-IgG, the caspase-3 expression level increased by 3.3 ± 2.21-fold, and, as expected, in shRNA-treated cells, its expression was reduced to 0.11 ± 0.03. Cells treated with camptothecin (CPT), an apoptotic inducer control, behaved similarly to those treated with PV-IgG, as the topoisomerase 1 poison increased caspase-3 expression to 2.92 ± 1.08. After inhibiting the caspase-3 mRNA, its transcription was no longer increased (Figure 3a). The specific α-tubulin-shRNA and the scrambled shRNA plasmid A used as negative controls did not prevent PV-IgG-dependent caspase-3 expression.

Western blot and immunofluorescence with anti-caspase-3 and anti-M30 monoclonal antibodies were performed; the latter tags neoepitopes in cytokeratin 18 targeted by active caspase-3. These techniques were used to indirectly assess the functional effect of the Casp-3-specific shRNA-induced knockdown, demonstrating that cells treated with PV-IgG fully expressed the caspase-3 protein. This expression was decreased by the Casp-3-shRNA, as shown in Figure 3b–d. We note that the effect of silencing caspase-3 by Casp-3-shRNA was complete in untreated cells and cells treated with N-IgG, but such an effect was partial in cells treated with PV-IgG and CPT.

### 2.4. Cell Viability

*HaCaT* cells were grown under different experimental conditions, as previously described. Cells without treatment reached 61% viability. Similar results were obtained with those incubated with N-IgG, which did not affect cell viability; by contrast, PV-IgG induced a decline in viability to ~46% (*p* < 0.0001), and this effect was reversed by pre-treatment with Casp-3-shRNA, which increased viability to ~60% (*p* < 0.0001). A positive control comprising apoptosis induced by CPT was used as a death control, decreasing cell viability to ~51%. Unexpectedly, pre-treatment with shRNA did not affect non-attached cell viability, indicating that the topoisomerase 1 poison possesses caspase-independent cell death effects (Figure 4a Viability).

### 2.5. Cell Adhesion

The polarity of each cell and its interaction with neighboring cells via the extracellular matrix are important for keratinocyte function, and the desmosome molecular complex maintains this property. The results show that desmosome-dependent junctions were disrupted by PV-IgG. The cell adherence under basal conditions without treatment was ~64 adherent cells/field, and this rate dropped to ~36 adherent cells/field with the PV-IgG autoantibody effect (*p* < 0.0004). This detachment effect was similar in cells treated with CPT; in both cases, pre-treatment with caspase-3-shRNA increased cell adherence (Figure 4b Adhesion).

### 2.6. Caspase-3-shRNA Reduces Cell Death Induced by PV-IgG

Under basal conditions, the rate of apoptotic *HaCaT* cells and cells incubated with N-IgG was ~21%, and the PV-IgG effect increased this to ~33.7% (*p* < 0.0001). As expected, the cells pre-treated with caspase-3-shRNA reduced cell death to ~20%, and the apoptosis was comparable to the basal condition. On the other hand, the CPT-positive control increased the apoptotic rate one-fold, and this effect was attenuated under the caspase-3-shRNA effect (Figure 4c Apoptosis).

### 2.7. Phosphatidylserine (PS) Exposure on the Cell Surface

The translocation of PS to the outer leaflet of the cell membrane is a sign of cells undergoing apoptosis; we used 6 h as the time at which apoptotic changes could be fully observed at the plasma membrane. As expected, cells under basal conditions without treatment and those treated with N-IgG did not exhibit relevant PS-Annexin staining: ~18 and 27.7, respectively. By contrast, the cells treated with PV-IgG displayed a strong green signal indicative of apoptosis (~124 (*p* < 0.0001)); this effect was reversed by caspase-3-shRNA. As expected, the control cells in which apoptosis was induced by camptothecin also showed an intense annexin-FITC signal: ~109. Interestingly, phosphatidyl serine exposure was prevented by suppressing caspase-3 mRNA (Figure 4d PS exposure). Taken together, our data demonstrate that PV-IgG reduces cell adhesion and viability and increases cell detachment. These data suggest that caspase-3 is linked to cell adhesion, since most of these effects were neutralized by the caspase-3-shRNA.

## 3. Discussion

In the present work, an in vitro shRNA approach was used against caspase-3 to demonstrate the possible role of this enzyme in acantholysis, the pathogenic hallmark of pemphigus. This tool was chosen because it can suppress mRNA targets, demonstrating the possible pathogenic participation of caspase-3 in pemphigus. This has been shown in the results of other investigators who silenced the desmoplakin and Dsg 3 genes and demonstrated the structural role of both desmosomal proteins in cell attachment, proliferation, and differentiation in human immortalized keratinocytes [15,16,17,18].

Since Anhalt et al. [19] demonstrated the pathogenic effect of blistering caused by PV-IgG, different pathophysiological models triggered by PV-IgG have been explored to determine the consequence of PV-IgG binding to Dsg1 and Dsg3. This issue is the subject of intense research, as different experimental and theoretical approach models have emerged to clarify the mechanism of acantholysis; presently, the most accepted notion is that the main mechanism of pemphigus vulgaris blistering is the direct steric interference of homophilic and heterophilic Dsg:Dsg and Dsc:Dsg binding caused by pemphigus autoantibodies [20,21,22]. This interference seems to imitate the so-called “tryptophan exchange” between interacting Dsg3 molecules, which affects its “cis and trans” interactions. Additionally, PV-IgG internalizes extradesmosomal Dsg3 through the p38MAPK and EGFR pathways, producing intracellular incoherence through cytoskeleton uncoupling and resulting in cell detachment [23].

We used *HaCaT* cells that mainly express Dsg 1 and Dsg 3 to explore the role of caspase-3 in cell adhesion. Our results demonstrated PV-IgG-induced cell detachment and apoptosis. The deleterious effect on keratinocyte survival induced by an excess of pemphigus autoantibodies was comparable to the effect induced by CPT, which is used as a control to induce apoptosis [24]. Our results coincide with other experimental studies that show some apoptotic features in acantholytic cells mediated by PV-IgG [25,26]. In this sense, we understand that these effects occur in vitro; consequently, they do not necessarily reflect what is happening in vivo in the tissues of pemphigus patients.

The link between pemphigus disease and cell death has been extensively studied but is still a controversial issue. Many observations have been made in pemphigus skin biopsies, and partial conclusions are usually based on detecting one or a limited number of apoptosis markers rather than extensive screenings of different cellular death pathways. Many studies, including our previous work, have not demonstrated caspase-3 pathway activation, and this raises the following question: is apoptosis relevant in pemphigus, or is it just an epiphenomenon? Presently, a large body of work on cell death in pemphigus is available, and interestingly, some studies are pro-apoptosis while others are anti-apoptosis. Thus, there is no conclusive verdict on this issue [7,8,9,25,26,27,28,29,30]. In this regard, Bumiller-Bini et al. conducted a meta-analysis, concluding that future and well-controlled studies on pemphigus and keratinocyte cell death apart from apoptosis are required [11]. This proposal is supported by the fact that there are at least 12 different cell death routes [31], and only two or three have been partially explored in pemphigus. Thus, it is obvious that conclusive remarks cannot be made in this regard. Another factor in this confusion is the ambivalent role caspase-3 plays in cell adhesion, survival, proliferation, and tumorigenesis, reported by different investigators [30,32,33]. Furthermore, the “non-lethal effects of caspase-3” on the adherent function of keratinocytes likely contribute to acantholysis, making everything more complicated. However, despite its limited contribution, our study provides the potential to expand the downstream markers that affect keratinocyte adhesion in PV. Epithelial cells in apoptosis have two stages that depend on the catalytic activity of caspase-3: the first one gives structure to focal adhesion by fragmenting adherent molecules; the second destroys the nuclear envelope through DNA and protein fragmentation [34,35]. Therefore, cleaving desmosomal proteins reduces the number of functional and active desmosomes and interferes with their de novo formation. This changes the keratinocyte shape and induces disorganization in intermediate filaments and molecules like β-catenin; the latter regulates cell adhesion affected by caspase-3. This is key in dissembling intermediate filaments. All of the above are involved in forming blisters in PV [36]. On the other hand, the non-catalytic effects that influence cell adhesion are regulated by the production of fibronectin and other extracellular matrix proteins directly involved in cell adhesion, proliferation, and survival. These issues depend on procaspase-3, regulating keratinocyte behavior in a pro-apoptotic environment [37]. These and other issues open up a range of potential factors related to adhesion, and we think that the contribution of our work will allow us to explore these factors in the future.

## 4. Materials and Methods

### 4.1. PV-IgG Characterization

A positive serum for anti-epithelial antibodies obtained from a female 28-year-old patient with pemphigus vulgaris was used for IgG purification. She had extensive blistering on the mouth and trunk, with a hematoxylin and eosin (H&E) skin biopsy showing intraepidermal blisters and acantholytic cells. Plasma obtained during therapeutic plasmapheresis was kindly donated to our lab and used to purify pemphigus IgG. The anti-epithelial reactivity of the pemphigus vulgaris serum was determined with indirect immunofluorescence (IIF) using monkey esophagus (Euroimmun, Medizinische Labordiagnostika AG, Lübeck, Germany) as an antigenic source and goat FITC-labeled polyclonal anti-human IgG as a secondary antibody (Sigma, St. Luis, MO, USA). After incubation and washing with PBS, the slides were evaluated under epifluorescence microscopy (Olympus BX40, Olympus, Co., Hachioji, Tokyo, Japan) [38]. Anti-desmoglein 1 and 3 antibody values were measured using ELISA (Euroimmun, Medizinische Labordiagnostika AG, Lübeck, Germany). In all assays, control human serum was included. All assays were performed in triplicate, and a negative control N-IgG was included. Patients were informed of the study and procedures and signed an informed consent form.

### 4.2. PV-IgG Purification

To start, plasma was used for gamma globulin precipitation with ammonium sulfate (Sigma-Aldrich, St. Louis, MO, USA). The precipitates were extensively dialyzed against distilled water using a cut-off membrane (MWCO) of 12 kDa. After complete salt removal, the precipitated gamma globulin was dialyzed against PBS; then, it was diluted in a binding buffer (0.02 M of sodium phosphate, pH 7.0). Gamma globulin was purified via affinity chromatography using a HiTrap^®^ Protein G prepacked column (GE Healthcare, Chicago, IL, USA). IgG bound to the G protein was eluted with 0.1 M glycine-HCl in pH 2.7 buffer. Then, the effluent was neutralized with 1 M of Tris-HCl buffer (pH 9.0) and dialyzed against PBS as previously described, stirred for 48 h at 4 °C. The fractions were characterized with 10% SDS-PAGE [39], and anti-epithelial activity was tested as previously described.

### 4.3. Cell Culture

The *HaCaT* human keratinocyte cell line was obtained from the American Type Culture Collection (CRL12191-ATCC, Manassas, VA, USA). It was grown in cell monolayers on well plates (Nunclon^TM^; Corning, NY, USA; Microscopy Chamber Ibidi; Thermo Fisher, Scientific, Waltham, MA, USA) and cultured in high-glucose DMEM (DMEM-HG, Gibco, Grand Island, NY, USA) supplemented with 10% FBS, 1× antibiotic–antifungal, 1× sodium pyruvate, and 1× L-glutamine (Gibco, Paisley, UK). The cells were incubated at 37 °C in a 5% CO_2_ atmosphere (Incubator NuAire, Plymouth, MN, USA). Endotoxin contamination was ruled out by the HEK-blue tm LPS detection kit 2 (InvivoGen, San Diego, CA, USA).

### 4.4. Cell Adhesion Assays

We used the attachment assay to detect the flattering of adherent cells under phase contrast microscopy [40]. These results were confirmed by colorimetrically detecting bound cells to an immobilized substrate using 0.1% (*w*/*v*) crystal violet; 200 mM of MES (4-morpholineethanesulfonic acid hydrate; 2-(N-Morpholino) ethane sulfonic acid hydrate) [41]. The viability of non-adherent cells was counted with a cell cytometer (see below).

### 4.5. Caspase-3-shRNA-Expressing Vector and Oligo Design for Simple Hairpin

We used the pGSH1-GFP vector (Gene Silencer^®^, 092205 MV) linearized by *BamHI* and *NotI* (Thermo Scientific, Waltham, MA, USA) and prepared for ligation. *Caspase-3* gene sequences (CASP-3-NCBI, accession number NM_032991.2) were selected using guided design software for shRNA synthesis (GenLink^TM^) (v1.2.1). shRNA was automatically designed by the software, and three sense and anti-sense sequences were obtained; however, only one of the three pairs, synthesized by a commercial company, was used at a concentration of 0.05 μg (Life Technologies, Invitrogen, Waltham, MA, USA). Sense: 5′-GATCCGAAAGCACTGGAATGACATCGAAGCTTGGATGTCATTCCAGTGCTTTTTTTTTGGAAGC-3′; anti-sense: 3′-GGCCGCTTCCAAAAAAAAAGCACTGGAATGACATCCAAGCTTCGATGTCATTCCAGTGCTTTCG-5′.

Several sequences were designed in the aforementioned software, of which three sets were chosen and synthesized. However, only one was used in all experiments, based on the following criteria: the presence of purine as the 5′-terminal nucleotide, ensuring the accuracy of start-site selection and boosting the transcriptional efficiency, and secondly, an shRNA stem of 19 bp, enabling competition between Dicer and Ago2 to process the shRNA, as is stated in Figure 2 [42].

Oligos were adjusted to reach a final concentration of 1 μg/μL and then annealed and ligated by T4 DNA ligase into a pGSH1-GFP vector. The ligase reaction mix was used for the transformation procedure with our own modifications. Then, *Casp-3*-shRNA pGSH1-GFP expression vectors were propagated into *DH5α* cells, achieving 92% efficiency. After propagation, the DNA was extracted, electrophoresed in 1% agarose gels in TAE 1×, stained with ethidium bromide, and run through a TAE buffer at 82 V for 1h. Positive shRNA-containing clones displayed linear plasmids on agarose gels, while negative clones appeared as supercoiled plasmids after *Hind III* digestion [43]. 

### 4.6. Transformation of DH5α E. coli (F− Φ80lacZΔM15 Δ(lacZYA-argF) U169 recA1 endA1 hsdR17 (rk−, mk+) phoA supE44 thi-1 gyrA96 relA1 λ−)

Casp-3-shRNA expression vectors were propagated into Smart Cells^TM^ chemically competent DH5α cells, which were mixed with the vectors at 42 °C for 45 s and then incubated in SOC medium at 37 °C for 1 h. Then, the transformation mix was spread on LB/agar plates containing 50 mg/mL kanamycin and incubated overnight at 37 °C. Positive transformants were selected from colonies via GFP expression under blue-light-emitting diode illumination.

### 4.7. Isolation of Casp-3-shRNA Expression Vectors from E. coli Transformants

Colonies were chosen, and recombinant plasmids were extracted via alkaline lysis with SDS and digested with *Hind III*. Miniprep DNA was run on a 1% ultrapure agarose gel. Positive shRNA-containing clones displaying linear plasmids on agarose gels were selected and grown overnight in LB media containing 50 mg/mL kanamycin to prepare sufficient quantities of plasmid DNA for transfection. Bacteria were incubated in an incubator/shaker overnight at 37 °C with constant shaking at 250 rpm and then collected and centrifuged at 5200× *g* for 5 min. Then, cells were treated with deep blue lysis buffer and neutralized with a clearing buffer. Samples were centrifuged in a zymo-spin, the Zyppy kit was washed, and the plasmid DNA was eluted with Tris-HCl (pH 8.5 and 0.1 mM EDTA) (Zyppy^TM^ Plasmid Miniprep Kit, Zymo Research, Irvine, CA, USA).

### 4.8. Transfection of Caspase-3-shRNA

Once the *HaCaT* cells were confluent at 90% cells per well, they were transfected with a cationic polymer transfection agent (jet-PEI, Polyplus Transfection): N/P = 5 and 4 µg of DNA (shRNA) in 100 µL of 150 mM NaCl. Cultures were made in triplicate and subjected to the following conditions: untreated—(1) negative control only in KSFM medium; (2) addition of purified IgG from normal human serum (N-IgG; 1 mg/mL); (3) addition of purified IgG from pemphigus vulgaris (PV-IgG; 1 mg/mL); (4) addition of Casp-3-shRNA+PV-IgG; (5) addition of positive apoptosis-inducing control with CPT (4 µg/mL) (C9911-Sigma, Saint Louis, MO, USA); (6) Casp-3-shRNA + CPT. Subsequently, 1 mL of high-glucose DMEM was added with an apoptosis inducer (CPT or PV-IgG), and the cells were incubated for 6 h at 37 °C in a 5% CO_2_ atmosphere.

### 4.9. Nonadherent Cell Viability Monitoring via Flow Cytometry

This assay was performed in a Guava Personal Cell Analysis Check System (Technologies PCA) as follows: a suspension of non-adherent cells was prepared by mixing 50 µL of the cells with 450 µL of Guava Via Count staining reagent (Cat. No. 4000-0040), and cell viability results were analyzed using CytoSoft 2.1.4.

### 4.10. Labeled Annexin Affinity Assay

Apoptotic PS exposure was examined in *HaCaT* cells as follows: The culture medium was removed from the wells with the specific treatment, and the cells were washed with PBS 1× pH 7.2 (Gibco^®^), fixed with 4% paraformaldehyde, and tagged with 100 µL of Annexin-V-Fluos labeling solution (Cat. No. 11 858 777 001, Annexin-V-Fluos Staining Kit, Roche Diagnostics, GmbH, Penzberg, Germany; http://www.roche-applied-science.com). After 10 min of incubation, apoptotic cells were analyzed with inverted fluorescence microscopy with an excitation range of 450–500 nm (Olympus IX71; DP71 camera; DP Manager 3.1.1.208 software). The PS areas were quantified in pixels using Image-Pro^®^ Plus vs. 7.0 (Media Cybernetics, Rockville, ML, USA). To distinguish living cells from apoptotic cells via microscopy, cells were counterstained with propidium iodide (red signal), and apoptotic PS membranes were detected using the Annexin-FITC reagent (Appendix A).

### 4.11. Caspase-3 Gene Expression Analysis via qRT-PCR

Total RNA was purified with a Pure Link^TM^ RNA mini-Kit (Ambion, Life Technologies, Waltham, MA, USA). Then, 1 µg of total RNA was added to a total reaction mixture of 25 µL with Super Script^TM^ One-Step RT-PCR and Platinum^®^ Taq (Invitrogen, Life Technologies, Waltham, MA, USA). The reactions were carried out in a Pixo Helixis system (Illumina) using Fast SYBR^®^ Master Mix (Applied Biosystems, Waltham, MA, USA) detection chemistry with a final volume of 20 µL. The samples were amplified at 95 °C for 20 s, followed by 40 cycles of denaturing for 3 s at 95 °C and annealing/extending for 30 s at 60 °C. The oligos used were caspase-3: sense: 5′-TTGTGGAATTGATGCGTGAT-3′ and anti-sense: 3′-GGCTCAGAAGCACACAAACA-5′ (accession number NM_032991.2). As a housekeeping control, the α-tubulin gene (sense: 5′-CTTCGTCTCCGCCATCAG-3′; anti-sense: 3′-TTGCCAATCTGGACACCA-5′) (accession number NM_006009.2) was amplified separately, and the experiments were performed in triplicate. The caspase-3 expression levels were normalized to that of α-tubulin, and each experiment was processed with its housekeeping control. α-tubulin-shRNA and scrambled-shRNA Plasmid-A (sc-108060, Santa Cruz, CA, USA) were used as negative controls. Relative gene expression levels were determined using the comparative CT method [44].

### 4.12. Caspase-3 Protein Expression

The indirect effect of shRNA on the expression of the caspase-3 protein was tested via Western blotting [45] using *HaCat* untransfected cell extracts treated with PV-IgG and transfected cells with Casp-3-sHRNA, followed by PV-IgG treatment. For this purpose, an anti-caspase-3 monoclonal antibody was used (caspase-3 (T46L):sc-56055, Santa Cruz Biotechnology, Dallas, TX, USA). Next, caspase-3 was cellularly localized using immunofluorescence with the described anti-caspase-3, followed by incubation with a secondary anti-mouse IgG-FITC labeled antibody. Another assay to confirm caspase-3 activity was carried out using the M30 CytoDEATH (Roche, Mannheim, Germany) mouse monoclonal antibody to detect apoptosis in epithelial cells (the caspase cleavage product of cytokeratin 18 that indirectly reflects caspase-3 activity) [46]. The intensity of the IIF and the cellular localization of caspase-3 and neoepitops of cytokeratin tagged by anti-M30 antibodies were analyzed in the software image-Pro Plus, Version 7.0 (Media Cybernetics, Rockville, MD, USA).

### 4.13. Statistical Analysis

The experiments were performed in triplicate, and treated and untreated samples were analyzed using two way-ANOVA, post-Tukey, and the Kruskal–Wallis post-Dunns test with GraphPad Prism version 9.2 (GraphPad, San Diego, CA, USA). *p* < 0.05 was considered statistically significant.

## 5. Conclusions

PV-IgG induces detachment of *HaCaT* cells and is a powerful trigger of caspase-3 transcription; such an effect is not induced by N-IgG. Specific Casp-3-shRNA decreases but does not stop the expression of caspase-3 induced by PV autoantibodies. The present results suggest that caspase-3 plays a role in the cell adhesion of immortalized keratinocytes, as shRNA prevents cell detachment and late apoptosis in *HaCaT* cells induced by PV-IgG. This effect is probably due to the non-lethal effects of Caspase-3. This furthers our understanding of the molecular role of caspase-3 cell adhesion dependence in pemphigus disease.

## Figures and Tables

**Figure 1 ijms-25-08864-f001:**
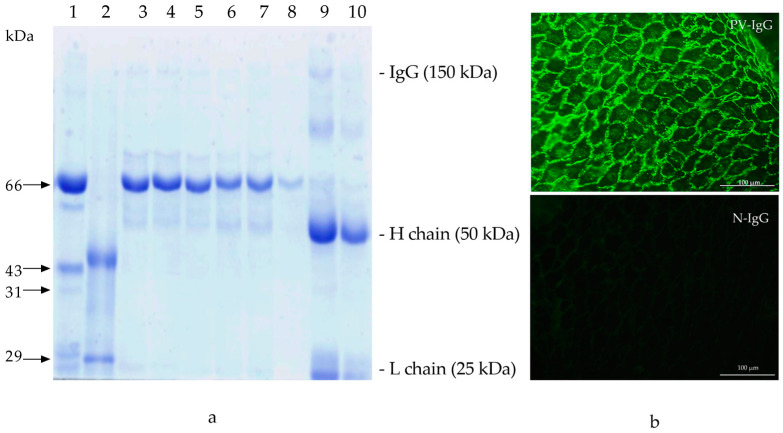
(**a**) PV-IgG showing anti-epithelial activity. (**a**) Purified IgG by PAGE. Lines: 1: MW1 (BSA 10 µg); 2: MW2 (Gibco^®^); 3–5: N-IgG (1:100, 1:200, 1:500); 6–8: PV-IgG (1:100, 1:200, 1:500); 9: N-IgG purified; 10: PV-IgG purified; (**b**) indirect immunofluorescence on monkey esophagus showing positive anti-epithelial activity of PV-IgG at a 1:2560 titer (40×); lower panel purified N-IgG showing negative anti-epithelial activity.

**Figure 2 ijms-25-08864-f002:**
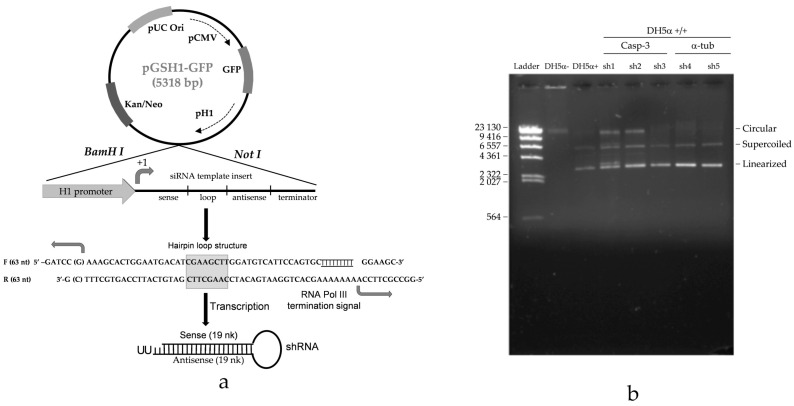
(**a**) The shRNA expression vector pGSH1-GFP and restriction sites map (Gene Silencer^®^, 092205 MV). The shRNA expression cassettes were cloned downstream of the H1 promoter, shown in black. This vector is based on a pUC vector backbone designed for high-efficiency propagation in Smart Cells^TM^ chemically competent *E. coli*. The bottom of the scheme shows the short hairpin used to selectively inhibit caspase-3, which was formed after sense and anti-sense sequences (middle) were linked in a loop form after binding each other, producing double-stranded RNA that could silence caspase-3 mRNA. (**b**) Panel plasmid DNA strains DH5α−, DH5α+, and DH5α+/+. Positive transformant clones containing caspase-3-shRNA displayed linear plasmids, while negative clones had a coiled shape.

**Figure 3 ijms-25-08864-f003:**
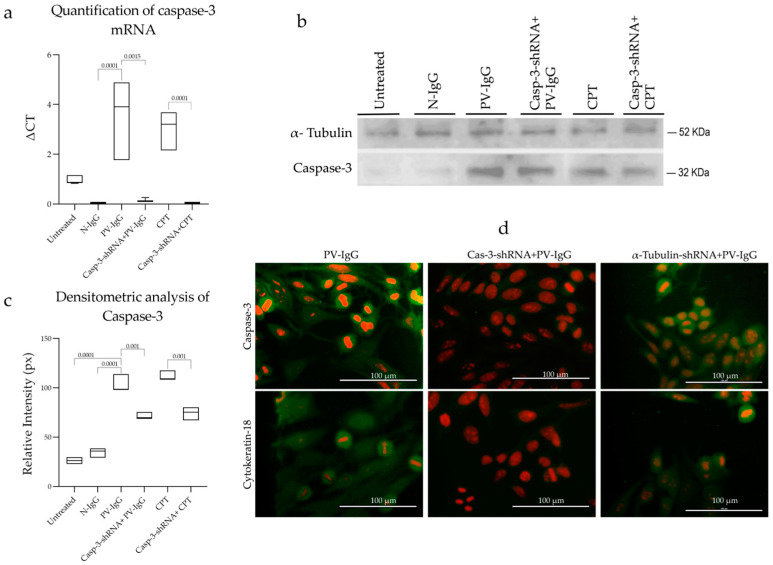
(**a**) qRT-PCR relative expression analysis of caspase-3 in untreated HaCaT cells and cells treated with N-IgG, PV-IgG, Casp-3-shRNA + PV-IgG, CPT, and Casp-3-shRNA + CPT. The treatments were incubated for 6 h until analysis. α-tubulin expression was used as an endogenous control. Cells treated with Casp-3-shRNA had a decreased caspase-3 expression. The test was run in triplicate, and the standard deviation is included in the graph (*p* < 0.05). (**b**) The caspase-3 processing graph reflects the densitometric value of pixels analyzed during the Western blotting (**c**) of untransfected HaCaT cells treated with PV-IgG. The absence of the caspase-3 band in untransfected cells and transfected cells treated with N-IgG and an intense 32 kDa band of caspase-3 in cells treated with PV-IgG are shown; such a band is reduced in cells transfected with Casp-3-shRNA and exposed to PV-IgG (*p* < 0.002). (**d**) Immunofluorescence of HaCat cell caspase-3 protein expression (in green); note that shRNA decreased in ~40% of caspase-3 expression. In the lower part of the panel, PV-IgG induces caspase-3 expression, which targets cytokeratin 18, resulting in neoepitopes that are tagged by anti-M30 antibodies. Thus, the inhibitory effect of shRNA was demonstrated by decreasing fluorescence. Cells counterstained with propidium iodide (red).

**Figure 4 ijms-25-08864-f004:**
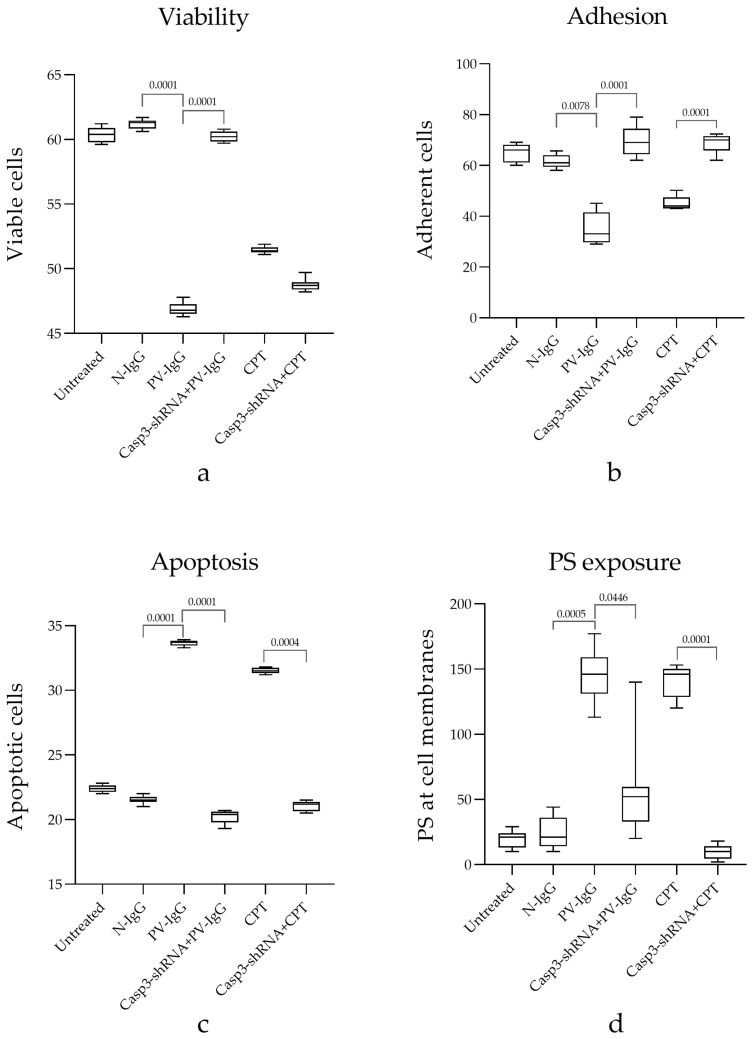
Graphs showing changes expressed by untreated *HaCaT* cells and cells treated with N-IgG, PV-IgG, Casp-3-shRNA + PV-IgG, CPT, and Casp-3-shRNA + CPT for (**a**) viability, (**b**) adhesion, (**c**) apoptosis, and (**d**) PS exposure. The significance threshold is *p* < 0.05. Adding PV-IgG significantly decreased the viability of HaCat cells, which detached from the culture dish. Apoptosis increased and, consequently, the mark of apoptotic membrane PS increased. Thus, the effect of Casp-3-shRNA is notable, neutralizing the effects of pemphigus autoantibodies.

## Data Availability

Data available upon request.

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
