# Peer review of "shRNA-Targeting Caspase-3 Inhibits Cell Detachment Induced by Pemphigus Vulgaris Autoantibodies in *HaCaT* Cells"

_ijms, 2024, doi:10.3390/ijms25168864_

Round 1
Reviewer 1 Report
Comments and Suggestions for Authors
The manuscript around Pacheco-Tovar et al. describes the effects of caspase 3 in HACAT-cells. To address this, they isolated PV igg from one patient as a mean of membrane dissociation inducer, knocked down caspase three, and looked at the effects such as apoptosis, cell adhesion and viability.
In general, I always find it troublesome to extrapolate results from one pv patient (untreated?) as general results for this aibd. Could you rather pool say IgG from 4-6 (male/female) PV patients (ideally untreated)? Optionally, use monoclonal antibodies such as AK23 (PMID: 21718682) and 2G4 (PMID: 36089007) and do some comparison in order to bring a bit more depth to the manuscript.
Second, the results need to be described more clearly, also in the legends (if (a) then there should be an (a) in the legend as well (see fig. 4).
Finally, the scientific value is generally quite limited. The effects of caspase 3 itself and its course in PV has been addressed quite extensively. Nonetheless, this approach here provides the potential to further pinpoint downstream markers affected in PV.
In normal confluent cells, there should be very little apoptosis. Judged from your suppl. fig 1a, your cells are heavily overcrowded. Please repeat to appr. 90% confluency and re-analyze.
Minor: please change the sentence l15-18
Results section: if you have (a), (b) please also refer in the text to those parts. Not just at the end as .Figure 2.
“This expression level was similar in cells treated with N-IgG” – it is reduced to nearly zero (??), also the abbreviation N-IgG is nowhere properly introduced
Please properly introduce the reason and function of CPT, now fig. 3 is hard to understand
L144 “PS exposure” ???
L140 “green” – colour is relative, therefore can be removed
è Please be more concise in the description of all the results
L19 Also”,” l25 In conclusion”,” …
L49 pls directly add a suitable ref at the end of the sentence
L55 and in general – please revise English
L206 IIF
M&M – for a company, include country
4.1.3 – I am sure you need to add that patient instruction (e.g. All participants were provided with oral and written information and signed a written consent…)
Fig. legend (a) vs. (A) (see supplementary fig.1)
Small personal remark: “cow's nose” this made me chuckle. At least in the European community I don t know anybody who has used that. At least as far as I know monkey oesophagus is the standard for IIF. But again, just a funny thought.
Comments on the Quality of English Language
please have it checked, especially for grammatical errors
Author Response
The English was professionally edited as requested by all reviewers.
- We clarified that the plasma corresponded to a patient with untreated pemphigus vulgaris, and we chose a single well-studied patient seen clinically, serologically and histopathologically, because it is easier to molecularly characterize a single biological sample and there is less possibility than other autoantibodies may produce cross-reaction and influence the results, Section 2.1 Line 88.
- The results guide in the “figure legends” has been expanded and is easily understandable for any reader.
- We totally agree with the reviewer about the limitations of the work, however he opens, as he very well cites, new horizons, regarding downstream markers that mediate the adhesion of keratinocytes in pemphigus.
- In material and methods, the experiment was repeated at 90% confluence as suggested by the reviewer and the density text of 5X105-2X106 was replaced by... at 90% per well, and a new set of photos where you can see the in supplementary figure 2.
- The suggestion to refer in Figure 2 as part a and b was appropriately noted in lines 96 and 98.
- In relation to the abbreviation N-IgG that was not introduced properly, it is noted at the beginning of the results as Normal IgG instead of Control IgG.
- The reason for including Camptothecin (CPT) in Figure 3 and throughout the manuscript this Topoisomerase I inhibitor was included as an apoptosis control and is noted in section 2.3, line 124.
- According to your suggestion green signal was removed. Section 4.10.
10, References were placed at the end of the corresponding paragraph.
11.IFI is IIF.
- Material and methods All commercial products mention their country of origin.
- A letter of consent signed by all participants who have reviewed and addressed the suggestions made by the reviewers is attached.
13.The patient was informed and signed his knowledge of participating in the study. Lines 278, 279 and 439, 440.
- Suplementary figures were properly labeled
- Indirect immunofluorescence was repeated in monkey esophagus instead of cow nose as suggested by the reviewer and Figure 1b was replaced.
Thank you very much in advance for the suggestions that were taken into account in their entirety, the revised manuscript is ready to be re-evaluated.

Reviewer 2 Report
Comments and Suggestions for Authors
In this study, the role of the pro-apoptotic protease, caspase-3, in the autoimmune disease pemphigus vulgaris was studied using a cell culture model. Autoantibodies were isolated from a patient and applied on normal cells and cells treated with shRNA against caspase-3. The authors conclude that the shRNA treatment inhibits cell detachment. The aim of the study is interesting, but the study design is not suitable.
As only short term effects were investigated, the relevance of the data for the pathology is questionable.
The mechanism of effects of caspase-3 on cell adhesion should be determined. The authors cite other papers that have reported evidence for caspase-3 contributions to pemphigus vulgaris. To make a significant advance over previous knowledge, more studies of molecular processes driven by caspase-3 are necessary.
It is not clear whether any control shRNA teatments (other than targeting caspase-3) were performed.
Investigations of caspase-3 protein (western blot, immunocytochemistry) are missing.
The catalytic activity of caspase-3 should be investigated.
HaCaT cells are a starting point for investigations, but primary keratinocytes should also be investigated.
The choice of CPT for comparisons needs to be explained and justified. The abbreviations should be explained at first use in the text.
Affinity purified IgG should be tested on human skin, not on cow nose, which is shown in figure 1b.
Figure 1b: The scale bar is missing. A negative control is missing.
The use of the English language needs to be improved.
Comments on the Quality of English Language
The use of the English language needs to be improved.
Author Response
Author’s reply to the review report (Reviewer 2):
- The reviewer rightly points out that the effects of PV-IgG are short-term, so the work data to understand the pathology are questionable. In this regard, we include a comment in the last two lines that indicate that the work simply explores a caspase-3-dependent adhesion pathway and we point out that our results do not necessarily reflect what happens in the tissues of patients with PV. On the other hand, another comment is made in the last paragraph of the introduction, it is pointed out that the effect of caspase-3 has been addressed mainly from the point of view of its enzymatic activity, rather than its effects related to non-lethal activity such as the case of cell adhesion, which is a mechanical aspect, and precisely with the focus of the present work we think that it will allow in the future to investigate the “downstream” adhesion markers affected by caspase-3 in PV.
- The only shRNA used was that of caspase-3
- In accordance with his suggestion, Western blot and immunofluorescence studies were carried out to detect the expression of the protein and the results are included in figure 3 and figure legend, lines 137 to 149. And in the material and methods section 4.12, lines 396 to 406.
- The catalytic activity of Caspase 3 was determined by Western blot, measuring the 19 kDa fragment Figure 3 (c) by densitometry, the values obtained were transformed to intensity per pixel and graphed in Figure 3 (b), while the presence of the Caspase 3 protein 3 was determined by IIF Figure 3 and is widely expressed in non-transfected cells but if treated with PV-IgG, said expression is suppressed by shRN; Fig 3 (d) and figure legends lines 137 to 149.
- We are aware and agree with the reviewer's observation that ideally we should have used more than one cell line, however we only used HaCaT cells, because they are immortalized keratinocytes that are relatively easy to maintain. The fact that we did not use other cells does not invalidate our results.
- The choice why camptothecin (CPT) was used is indicated in paragraph 2.3, line 124 and 128 to 132.
- The suggestion that the purified IgG should be tested on human skin and not on a cow's nose, the alternative was taken to test it on monkey esophagus, with normal human skin, so Figure 1b was replaced.
- Bar scale was included in Figures 1. (b) and Figure 3. (d).
In advance we thank the reviewer for his valuable observations and suggestions, all of them were taken into account, and we feel that the manuscript has improved and is ready to be reevaluated.

Reviewer 3 Report
Comments and Suggestions for Authors
The manuscript entitled “shRNA targeting caspase-3 inhibits cell detachment induced by pemphigus vulgaris autoantibodies in HaCaT cells” by Deyanira Pacheco-Tovar et al., was presented as an Original Article belonging to the “Molecular Biology” section (Special Issue: Molecular and Cellular Mechanisms of Skin Diseases). From the title the authors aimed to provide evidence about the effects of Caspase-3 silencing on keratinocytes cell adhesion and apoptosis by induced by PV-IgG. Although the thematic of the research is relevant and the article could be of potential interest not only for a wide community of basic researchers working on skin diseases but also for clinicians and dermatologists, in the current state the manuscript appears still preliminary, and the quality of the data does not reach the expected quality standards required for publication. Several major points need to be addressed to strengthen the technical rigor.
Comments for the text:
Globally, the manuscript needs to be improved in terms of data presentation and writing to make the work more accessible and complete to the reader. The introduction needs to be revised: the authors should provide a more detailed background and highlight the objectives, aims and novelty about this study.
Comments for the data:
In general, the data presented in the figures need to be improved. One main limitation of the study is that the validation methods are not abundant enough. Therefore, to support the conclusions, the results should be improved including several experimental approaches and different methodologies.
1. The authors presented the paragraph “2.2. Casp-3-shRNA expression in transformants” as a Results section. The reviewer retains that it should be directly integrated as a complementation of Material and Methods “4.5. Caspase-3-shRNA expressing vector and oligo design for simple hairpin” since does not add any informative results to the data presentation.
2. To assure that caspase-3 expression was inhibited, the authors evaluated the effect of the short hairpin on caspase-3 by qRT-PCR assay. This is not satisfactory and sufficient. To validate the effective optimized and functional effects of a stable short hairpin, it is necessary to examine the effects of a shRNA expression in term of protein expression by Western Blot.
3. It is commonly known that to avoid off-target effects, a minimum of three target sequences should be designed and tested for each gene, in order to increase the likelihood that at least one sequence results in significant gene knockdown. In the Material and Methods section “4.5. Caspase-3-shRNA expressing vector and oligo design for simple hairpin”, the authors have stated that “The shRNA were automatically designed by the software and three sense and anti-sense sequences were obtained, but only one was synthesized by a commercial company at concentration of 0.05 μg”. Based on what criteria did they choose one couple of sense/antisense sequences rather than the other two?
4. The qRT-PCR relative expression analysis of caspase-3 in HaCaT cells (Figure 3) was normalized against a single housekeeping gene, which is somewhat unusual. Have the authors tested other housekeeping genes to confirm stability of α-tubulin under the treatment conditions?
5. The authors analyzed the effect of Caspase-3 silencing on cell viability, cell adhesion and apoptosis. However, the methods used, and the presented results are not sufficient to support the conclusions. Other experiments are needed to sustain the authors’ hypothesis. Cell survival data should be improved using other methodologies, such as 5-Ethynyl-2'-deoxyuridine (EdU) assay, Cell-Cycle analysis or mRNA/protein expression analysis of proliferative markers (i.e. Ki67 or Cyclins). Analogously, the effects of the Caspase-3 silencing on cell apoptosis should be further analyzed evaluating mRNA/protein expression of key anti- and pro-apoptotic markers (i.e. Bcl-2, Bax, or Caspases).
6. Using just one cell line in the study is inadequate. It's recommended to incorporate a minimum of two different cell lines for more comprehensive results in the cell line-related experiments, especially because it is a study exclusively in vitro.
Comments for the figures:
1. Figures and Supplemental Figures are not properly cited in the Results section. Make sure to correctly cite each figure in the text, not only as a whole figure, but also as individual panel A/B/C/D and so on.
2. Figure 1B: The immunofluorescence staining needs to be better represented and described. To facilitate the interpretation of the presented data, the authors should indicate next to each immunofluorescence panel or inside each immunofluorescence box the name of the antibodies they are stained for, with a specific color-code (green/red). Improve the quality of the images. Show a zoom (63x) of each representative field. Scale bar and magnification (x) are missing. Please specify them in the relative Figure Legend.
Author Response
Author’s reply to the review report-Reviewer 3
The reviewer rightly points out that in the introduction a justification must be provided that indicates the novelty of this study and therefore its justification for publication. To improve this aspect, we point out in the last paragraph of the introduction that the effect of caspase-3 has focused mostly on its final effects on cell death, but little on cell adhesion, this is important since the loss of adhesion Cellular is a key factor that from a mechanical point of view is crucial in the formation of the blister, which is the clinical and histopathological characteristic of pemphigus, this is pointed out at the introduction section, lines 40 to 57.
- The reviewer suggests moving paragraph 2.2 casp-3-shRNA expression in transformants to material and methods section 4.5. This has been done.
- The reviewer suggests performing additional assays to qRT-PCR that demonstrate that caspase-3 expression is inhibited by Casp-3 shRNA. For this purpose, two tests were carried out: One to evaluate the effect of shRNA on the expression of the Caspase-3 protein by Western blot. Second: cellular localization by immunofluorescence with an anti-Caspase-3 monoclonal antibody. By both methods, the Caspase-3 protein was demonstrated in untransfected cells and treated with PV-IgG, however this expression of caspase-3 in cells transfected with sHRNA was significantly reduced. Another assay was performed with the M30 antibody that recognizes a neoepitope on cytokeratin 18 (J Cell Biol 148(6)2000 1239-54), confirming that caspase-3 was activeblot and immunofluorescence. Lines 29, section 2.3 lines 128 to 132. Figure 3 (modified) and figure legend, lines 135 to 149.
- The reviewer asks about the criteria we followed for the synthesis of sense and antisense sequences. As noted in section 4.5, several sequences were designed in the aforementioned software, of which three sets were chosen and the three sets were synthesized, however only one was chosen using the following criteria: The sequence that produced the highest percentage of transfection and more ease of selection of shRNA positive cells, another selection criterion was to avoid the presence of more than four A's or T's in the target sequences to avoid the problem of premature transcription termination. Another criterion is the presence of more than 50% of cytokines and guanines in the sequence of the oligo chosen for greater stability and affinity (Methods Mol Biol. 2010; 629: 141–158.). We clarify that three sets of sequences were synthesized and we chose a single sequence to perform the experiments. Section 4.5 lines 312 to 313; 319 to 323.
- We clarify that we did not use another gene other than alpha-tubulin as a control for the qRT.PCR.
- The reviewer suggested studying the expression of other markers, however, due to time constraints, we were only able to study the M30 marker of caspase-3 activity, as previously stated. Section 4.12 lines 396 to 406.
We thanks for your suggestions and observations that were taken into account and we feel the improved manuscript is ready for evaluation.

Round 2
Reviewer 1 Report
Comments and Suggestions for Authors
Rev 1 –
Dear authors, thank you for further improving the manuscript.
There are unfortunately still points that I find troublesome. You added this huge junk of text l40. But this is of no interest at all in this paper. You don t further discuss this, so why did you add it?
Also, there is no “Pemphigus foliaceous”, at least not in englisch
L90 – “Normal IgG (N-IgG) was negative for anti-epithelial and anti-desmoglein 1 and 3 activities (Figure 1 a and b)” Where do you show the visualization of that sentence?
Figure legends in general – please have look at
https://www.frontiersin.org/journals/immunology/articles/10.3389/fimmu.2024.1398120/full
Ideally, a very brief headline. Labelling of each sub-figure top left, statistics etc at the end of the legend. Exact M&M belong in the M&Ms (see l113, unnecessary)
Casp3 vs Casp-3 vs Cas3
Fig. 3C – it would be manageable to do a proper WB were 2 bands a visible, no?
Fig.3 – So you basically knock out Casp-3 (a), but you still have about halv the activity left? How do you explain that?
Fig. 3 legend “they show an intense 16 kDa band of active caspase fragments compared with cells transfected with Casp-3-shRNA and then exposed to PV-IgG, which show a predominant intact 32 kDa band.” – to really prove (a) one would expect a full WB showing Casp-3 in all the samples. Same goes for (d)
L123 – “caspase-3 transcription reduced its expression to 0.11 ± 124 0.03” pls rewrite that sentence
L129 - the short hairpin – of the Casp-3 specific shRNA induced knockdown
L228 “in this sense, we understand that these effects are in vitro” – like this you write in the communication, not in the paper
4.1 – so here you actually use three ways to refer to materials. Name/country, name/town/county, only name – as in every proper paper à name, town, country!! (if introduced first in the paper)
“Several sequences were designed in the aforementioned software, of which three sets were chosen and synthesized; however, only one was used in all experiments, based on the following criteria: it induced less toxicity in cells, produced the highest transfection percentage, and eased selecting shRNA-positive cells.”
This doesn t have any scientific value. Or show the actual data if you think it is important that your shRNA is only slightly toxic
Did you look on LPS contamination? (e.g. LAL assay?)
Comments on the Quality of English Languageok
Author Response
Author’s reply to the review report-Reviewer 1 Round 2
Reviewer 1 –
Dear authors, thank you for further improving the manuscript.
Query. There are unfortunately still points that I find troublesome. You added this huge junk of text l40. But this is of no interest at all in this paper. You don t further discuss this, so why did you add it?
Response: We agree with this observation, for this reason the clinical classification was omitted, and part of this paragraph was rephrased lines 40 to 42. There are different clinical forms of pemphigus which histologically correlate with the molecular and structural expression of Dsg1 or Dsg3 in epidermis and mucous membranes (lines 40-43).
Query. L90 – “Normal IgG (N-IgG) was negative for anti-epithelial and anti-desmoglein 1 and 3 activities (Figure 1 a and b)” Where do you show the visualization of that sentence?
Response. Figure 1 was reformed and a negative control figure was included, and the sentence of this line can be visualized at figure 1c and legend of figure 1. Lines 86-87.
Query: The reviewer suggest to have a look the article fimmu.2024.1398120 figure legends to have a better idea to improve the labelling of each sub-figure, to avoid the unnecessary M&M description (see L113 unnecessary).
Response: We agree with the unnecessary M&M description, L 113 was changed to 4.5 section third paragraph, line 322 to 323.
Query: Casp3 vs Casp-3 vs Cas3
Response: Nomenclature of caspase 3 was homogenized along the manuscript.
Query: it would be manageable to do a proper WB were 2 bands a visible, no?
Response: A new Wb under different experimental conditions was carried out including a tubulin as a house keeping protein. Figure 3 b. We pointed out that Cas-3-shRNA treatment reduces in 40% protein expression. Lines 138-139.
Query: Fig.3 – So you basically knock out Casp-3 (a), but you still have about halv the activity left? How do you explain that?
Response: We agree with your observation, cell transfected with Casp-3-shRNA abrogate Caspase-3 expression in normal cells and in those treated with N-IgG, however Casp-3 transcription is incomplete silenced and is activity is decrease in 40% only. Lines 137-139 and lines 415 to 417.
Query: L 123 – “caspase-3 transcription reduced its expression to 0.11 ± 124 0.03” pls rewrite that sentence
Response: In section 2.3 all paragraph was modified as follows (line 107 to 110: shRNA reduced caspase-3 mRNA expression, and under basal conditions, was ~1 ± 0.21-fold and this level was similar in cells treated with N-IgG; however, with PV-IgG, the caspase-3 expression level increased by 3.3 ± 2.21-fold, and as expected that shRNA-treated cells reduced its expression to 0.11 ± 0.03. Cells treated with camptothecin (CPT), an apoptotic inducer control, behaved similarly to those treated with PV-IgG, as the topoisomerase 1 poison increased caspase-3 expression to 2.92 ± 1.08. After inhibiting the caspase-3 mRNA, its transcription was no longer increased (Figure 3.a).
Query: L129 - the short hairpin – of the Casp-3 specific shRNA induced knockdown
Response: The phrase was changed as suggested by…. Casp-3 specific shRNA induced partial knockdown. Line 119.
Query: L228 “in this sense, we understand that these effects are in vitro” – like this you write in the communication, not in the paper
Response: We slightly modify the paper, clarifying that the effects were in vitro, this is highlighted in red along manuscript.
Query: 4.1 – so here you actually use three ways to refer to materials. Name/country, name/town/county, only name – as in every proper paper à name, town, country!! (if introduced first in the paper)
Response: Procedence of materials was homogenized as you suggested
Query: “Several sequences were designed in the aforementioned software, of which three sets were chosen and synthesized; however, only one was used in all experiments, based on the following criteria: it induced less toxicity in cells, produced the highest transfection percentage, and eased selecting shRNA-positive cells.”
This doesnot have any scientific value. Or show the actual data if you think it is important that your shRNA is only slightly toxic
Response: Thanks for your observation, we clarify in section 4.5, lines 312 to 317.-- that the criteria were: the presence of a purine as the 5’-terminal nucleotide that ensures the accuracy of start-site selection and boosts transcriptional efficiency, and secondlyas is stated in figure 2, the stem of our shRNA was 19bp, to be able for Dicer and Ago2 compete to process the shRNA, also the reference 41 was updated.
Query: Did you look on LPS contamination? (e.g. LAL assay?)
Response: We clarify in section 4.3 that endotoxin contamination was ruled out in specific assay. Lines 291-292.

Reviewer 2 Report
Comments and Suggestions for Authors
The manuscript has been improved, but there are still major weaknesses. The scope and relevance of the article are summarized in the my comments on the first version of the manuscript. The revised version is still interesting, but is lacks important controls. Additional experiments are necessary.
Specific comments:
In contrast to the statement “4. The catalytic activity of Caspase 3 was determined…” in the response to the reviewer’s comments, the catalytic activity of caspase-3 was NOT determined in this study. Indirect data were obtained, but activity was not measured.
Figure 3b: The detection of cleavage of caspase-3 is only an indirect and uncertain measure of caspase-3 activity. Therefore, the title of this panel should be “Cleavage” or “Processing” of caspase-3, but not “Activity”. The authors have not measured activity.
The western blot does not really show a band at the size expected for catalytically active caspase-3. There is smear which may included the predicted size of mature caspase-3, but this is not validated.
“M30” needs to be explained clearly in the legend. The labeling in the figures is not logical, because “caspase-3” is an antigen and “M30” is an antibody.
Figure 3d: Explain the colors.
In their response to the reviewer’s comments, the authors state “2. The only shRNA used was that of caspase-3”. Without control shRNA (scrambled sequence or another shRNA for comparison), the conclusions are not valid.
The authors do not report about negative controls for immunofluorescence, but these controls are essential.
Lines 289-291: “The fractions obtained were characterized by 10% SDS-PAGE [38] and activity anti-epithelial was tested as previously described.” The references is missing. Again, there is no mention of any control experiments such as only second step antibody for IF.
Comments on the Quality of English LanguageThe English language is used well.
Author Response
Author’s reply to the review report-Reviewer 2 Round 2
Query: …reviewer’s comments, the catalytic activity of caspase-3 was NOT determined in this study. Indirect data were obtained, but activity was not measured.
Response: We clarify in the headline 4.12 section is “Caspase-3 protein expression” Lines 391-392 and 401 rather than activation, and along this paragraph is clarified that anti-M30 antibody detect neoepitops induced by caspase-3 activity and indirectly reflects activation. Also this observation was clarified at figure legend 3. Lines 140-142.
Query. Figure 3b: The detection of cleavage of caspase-3 is only an indirect and uncertain measure of caspase-3 activity. Therefore, the title of this panel should be “Cleavage” or “Processing” of caspase-3, but not “Activity”. The authors have not measured activity.
Response: As reviewer suggest the term “activity” of figure 3 was changed by “processing”, it was changed, line 134.
Query: The Western blot does not really show a band at the size expected for catalytically active caspase-3. There is smear which may included the predicted size of mature caspase-3, but this is not validated.
Response: We agree with your observation for this reason new western blot assay was carried out that demonstrate a 32 kD band in PV-IgG and CPT treated cells, as such expression was decreased under Casp-3-shRNP effect, so this means that the silencing effect was insufficient as pointed out in lines 119 to 124.
Query: M30” needs to be explained clearly in the legend. The labeling in the figures is not logical, because “caspase-3” is an antigen and “M30” is an antibody.
Response: We agree with this comment and figure labeling of 3(d) as well as figure legends were reformed and is more logical. Lines 139 to 143.
Query: Figure 3d: Explain the colors.
Response: Figure 3(d), legends were explained. Lines 142-143.
Query: In their response to the reviewer’s comments, the authors state “2. The only shRNA used was that of caspase-3”. Without control shRNA (scrambled sequence or another shRNA for comparison), the conclusions are not valid.
Response: Addittional experiments including the constitutive alpha tubulin constitutive gen and the scrambled shRNA were used as negative controls and is stated in lines 112 to 115.
Query: The authors do not report about negative controls for immunofluorescence, but these controls are essential.
Response: Negative controls for immunofluoresceence were included in figures 1 and 3.
Query. Lines 289-291: “The fractions obtained were characterized by 10% SDS-PAGE [38] and activity anti-epithelial was tested as previously described.” The references is missing. Again, there is no mention of any control experiments such as only second step antibody for IF
Response: A figure of negative control was included for immunofluorescence in Figure 1(b), also at figure legend. Equally in section 4.1, line 269-270, it was stated that Normal-IgG was included as control. Also reference [39] for anti-epithelial activity by immunofluorescence was included. The reference of epitelial activity was included as reference 38 Line 535. Also a negative control was includede in Figure 1b.

Reviewer 3 Report
Comments and Suggestions for Authors
Although the authors tried to answer to the reviewer’s requests, no all experimental points were solved. Nevertheless, the reviewer appreciates the authors' efforts to address her previous comments and, finding the revised version of the manuscript improved in term of data presentation and organization, approves the publication of the manuscript in the current state.
Author Response
Dear Reviewer 3:
We appreciate your comments and suggestions in this revised version of the second round where experimental and supplementary data were incorporated throughout the manuscript and are highlighted in red.
Basically, the results of caspase 3 and constitutive alpha-tubulin protein expression were incorporated in Figure 3 and the legend to Figure 3, lines 119-123 and lines 129-142.
Furthermore, it is observed that α-tubulin shRNA and scrambled shRNA plasmid-A were well transfected into HaCaT cells and were used as negative controls, lines 387 and 388 and lines 113-114.
Furthermore in the conclusion section, lines 415-417, it was stated that PV-IgG induced caspase-3-dependent HaCaT cell detachment and casp-3-shRNA decreased but did not stop the expresion of caspase-3 triggered by PV-IgG. Lines 415-418.
We also clarify that endotoxin contamination was ruled out through a specific test; lines 291-292.
Finally we believe that all suggestions were addressed.
Round 3
Reviewer 2 Report
Comments and Suggestions for Authors
The revision has improved the quality of the manuscript.